# CoCoGesture: Towards Coherent Co-speech 3D Gesture Generation in the Wild

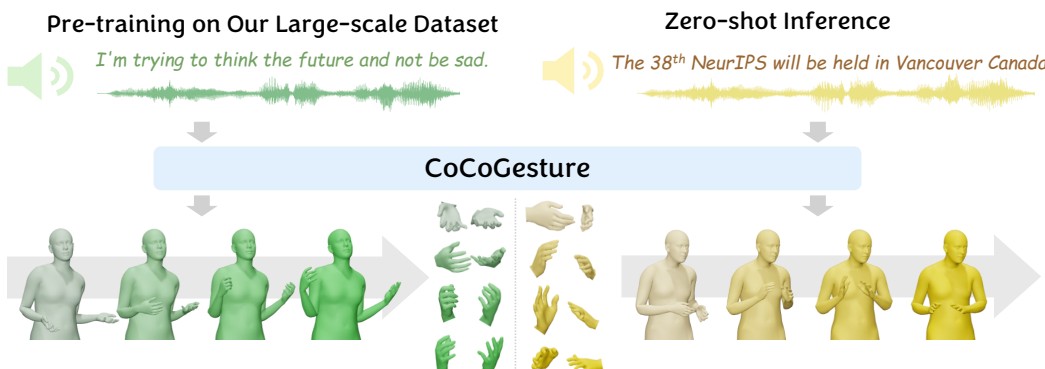

Figure 1: Our CoCoGesture framework pre-trained on the large-scale dataset can generate coherent and diverse 3D co-speech gestures corresponding with unseen zero-shot human audios.

## Abstract

Deriving co-speech 3D gestures has seen tremendous progress in virtual avatar animation. Yet, the existing methods often produce stiff and unreasonable gestures with unseen human speech inputs due to the limited 3D speech-gesture data. In this paper, we propose **CoCoGesture**, a novel framework enabling coherent and diverse gesture synthesis from unseen human speech prompts. Our key insight is built upon the custom-designed pretrain-fintune training paradigm. At the pretraining stage, we aim to formulate a large generalizable gesture diffusion model by learning the abundant postures manifold. Therefore, to alleviate the scarcity of 3D data, we first construct a large-scale co-speech 3D gesture dataset containing more than $40M$ meshed posture instances across $4.3K$ speakers, dubbed **GES-X**. Then, we scale up the large unconditional diffusion model to 1B parameters and pre-train it to be our gesture experts. At the finetune stage, we present the audio ControlNet that incorporates the human voice as condition prompts to guide the gesture generation. Here, we construct the audio ControlNet through a trainable copy of our pre-trained diffusion model. Moreover, we design a novel Mixture-of-Gesture-Experts (MoGE) block to adaptively fuse the audio embedding from the human speech and the gesture features from the pre-trained gesture experts with a routing mechanism. Such an effective manner ensures audio embedding is temporal coordinated with motion features while preserving the vivid and diverse gesture generation. Extensive experiments demonstrate that our proposed CoCoGesture outperforms the state-of-the-art methods on the zero-shot speech-to-gesture generation. The dataset will be publicly available at: *https://anonymous.4open.science/w/GES-X/* .

## 1 Introduction

Co-speech gesture generation aims to synthesize vivid and diverse human postures coordinated with the input speech audio. These non-verbal body languages greatly enhance the delivery of speech content in daily conversations (Qi et al., 2024; 2023a; Liu et al., 2024a). Meanwhile, synthesizing co-speech gestures of human avatars plays a significant role in wide applications like robotics (Farouk, 2022), virtual/augmented reality (AR/VR) (Fu et al., 2022), and human-machine interaction (Koppula & Saxena, 2013; Liu et al., 2023a).

Conventionally, recent researchers deal with speech-to-gesture tasks by modeling human upper-body dynamics with consistent speech voice (Liu et al., 2024a; 2022a; Yi et al., 2023; Chen et al., 2024; Liu et al., 2024b; Qi et al., 2024). Most of them address this task by conducting end-to-end mapping through the pre-defined corpus (Liu et al., 2022a; 2024a; Yi et al., 2023). However, they usually heavily rely on the paired audio-gesture data covering limited speaker identities, resulting in insufficient diversity of gestures. Moreover, the narrowed corpus data may lead to the model falling short of generalizing to unseen out-of-domain audio inputs, as shown in Figure 2(a).

In this work, we introduce the task of coherent and diverse co-speech 3D gesture generation from in-the-wild human voices, depicted in Figure 1. To achieve this goal, there are two main challenges: 1) The existing 3D meshed co-speech gesture datasets (Liu et al., 2024a; Yi et al., 2023) are insufficient to train a generalizable model. Creating such a dataset through accurate motion capture systems is extensively labor-consuming. 2) Modeling the coherent and diverse co-speech gestures from unseen human audio in an end-to-end fashion is difficult, especially in long sequences.

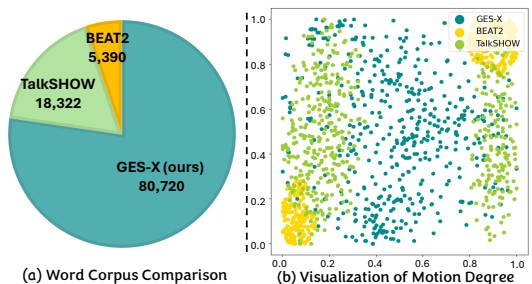

Figure 2: Dataset statistical comparison between our GES-X and existing meshed co-speech gesture datasets (*i.e.*BEAT2 (Liu et al., 2022a), TalkSHOW (Yi et al., 2023)). Our GES-X has a much larger word corpus and a more widely distributed gesture motion.

To overcome the issue of data scarcity, we first newly construct a large-scale 3D meshed co-speech whole-body dataset that contains more than 40M posture instances across about 4.3K aligned speaker audios, dubbed **GES-X**. Specifically, thanks to the advanced pose estimator (Zhang et al., 2023a), we can obtain high-quality 3D postures (*i.e.*, SMPL-X (Pavlakos et al., 2019) and FLAME (Li et al., 2017)) from in-the-wild talk show videos. Then, by employing WhisperX (Bain et al., 2023) for automatic speech recognition, we ensure the acquired text transcript and phoneme consistency with speaker audio. In this fashion, our GES-X provides the most comprehensive co-speech gestures with diverse modalities. As reported in Figure 2 (b), the posture motion degree of the GES-X dataset displays a much more widely uniform distribution against others, indicating our dataset contains more diverse gestures. Meanwhile, the common mesh standards in our dataset also support other downstream human dynamics-related tasks, *e.g.*, talking head generation (Tian et al., 2024), human motion generation (Ao et al., 2023).

Along with this dataset, we propose **CoCoGesture**, a novel framework that enables the generation of coherent human gestures from the unseen voice. Our key insight is built upon the custom-designed pretrain-fintune training paradigm. To ensure the generalization of the pre-trained model, we leverage our large-scale co-speech gesture dataset GES-X as the source training set. Specifically, we first conduct the pre-training phase based on the large unconditional diffusion transformer backbone (Peebles & Xie, 2023). This diffusion model serves as a gesture expert and is scaled up to 1B parameters, thereby enabling the training model to build the sufficiently inherent motion manifold from massive gesture dynamics. In this manner, our pre-trained model ensures the realism of the generated gestures while preserving vividness and diversity.

Moreover, to incorporate the human speech as the conditional prompt coordinately, inspired by (Zhang et al., 2023b), we present the audio ControlNet for fine-tuning. Concretely, we refactor a trainable copy of our pre-trained unconditional large model for adapting various audio conditions. Then, we propose a novel block, named Mixture-of-Gesture-Experts (MoGE), to fuse the audio embedding from the human voice and the gesture features from pre-trained gesture experts through a routing mechanism. Here, the routing mechanism adaptively balances the input audio signal features with the retained original motion clues. Meanwhile, the learned temporal-wise soft weight of the routing mechanism greatly guarantees generated results to maintain the coherence rhythm with input human speeches. Extensive experiments conducted on the out-of-domain datasets (Liu et al., 2024a; Yi et al., 2023) demonstrate our fine-tuned framework synthesizes vivid and diverse co-speech gestures, outperforming the state-of-the-art counterparts. Our GES-X dataset will be open-sourced soon to facilitate the research on the relevant community.

Overall, our contributions are summarized as follows:

- We introduce the task of co-speech gesture generation from in-the-wild human speech incorporating the large 3D meshed whole-body human posture dataset, named GES-X. It includes more than 40M high-quality gesture instances with 4.3K speakers, significantly facilitating research on diverse gesture generation.

- We propose a novel framework named CoCoGesture that leverages the Mixture-of-Gesture-Experts (MoGE) blocks to adapt various unseen audio signals with pre-trained highly generalizable gesture experts effectively. The presented MoGE greatly enhances the temporal coherence between generated results and conditional prompts.

- Extensive experiments show that our CoCoGesture produces vivid and diverse co-speech gestures given unseen human voices, outperforming state-of-the-art counterparts.

## 2 RELATED WORK

**Co-speech Gesture Generation.** Generating vivid and diverse co-speech gestures has witnessed impressive progress in recent years due to its practical value in wide-range applications (Qi et al., 2023c; Liu et al., 2023a; Zhu et al., 2023; Liang et al., 2024; Tian et al., 2024). Conventionally, researchers utilize the rule-based workflow to bridge the gap between human speech and gestures via the pre-defined corpus by linguistic experts (Marsella et al., 2013; Poggi et al., 2005). Other works generate the results relying on mapping the audio signals to manually defined gesture features through machine learning (Cassell et al., 1994; Huang & Mutlu, 2012). Nevertheless, these two approaches both need much more effort in preliminary dataset design, causing them to be limited by the size and quality of the datasets.

Recently, thanks to the advanced deep learning methods and 3D human body modeling techniques (Loper et al., 2023; Zhang et al., 2023a; Pavlakos et al., 2019; Boukhayma et al., 2019; Li et al., 2017), many works are proposed to generate the continuous 3D upper body postures. Speech-gesture-aligned datasets (Liu et al., 2022b; Yi et al., 2023; Yoon et al., 2020; Liu et al., 2024a; 2022a) are also proposed to address this challenging task. They involve multi-modality clues to promote the generated gestures to be much more reasonable and diverse, like emotion (Liu et al., 2022a; Qi et al., 2023a; 2024; Bhattacharya et al., 2021), identity (Yi et al., 2023; Liu et al., 2022b; 2024b), text transcript (Liu et al., 2022b; Cheng et al., 2024). To be specific, Ao *et. al* (Ao et al., 2022)propose a rhythm-based segmentation pipeline to boost the harmony between speech and gestures. Yang *et. al* (Yang et al., 2023) leverage emotion as guidance to produce various stylized gestures with the specifically designed diffusion model. Ahuja *et. al* (Ahuja et al., 2020) mix the disentangled gesture styles as an ensemble to guide the gesture generation. However, they overlook that directly generating the gesture from an in-the-wild human voice is much more practical in real-world scenes. Considering the previous datasets are restricted to a limited scale, we thus propose a large-scale meshed 3D co-speech dataset to facilitate the research on audio-driven gesture generation from in-the-wild human speeches.

**Zero-shot Human Motion Generation.** Human motion generation strives to generate natural sequences of human poses. Recent advancements in motion data collection and generation methods have sparked growing interest in this field. Existing research primarily revolves around generating human motions using conditional signals like text (Tevet et al., 2022b; Chen et al., 2023; Dabral et al., 2023), audio (Tseng et al., 2022; Ao et al., 2023; Zhu et al., 2023), and scene contexts (Araujo et al., 2023; Huang et al., 2023). Currently, open-set human motion generation focuses on zero-shot text-driven generation (Reed et al., 2016; Lin et al., 2023), which creates new content from text prompts without relying on pre-defined data. MotionCLIP (Tevet et al., 2022a) enhances zero-shot generation by employing a Transformer-based autoencoder to align the motion manifold with the latent space of pre-trained vision-language model CLIP (Radford et al., 2021). However, without sufficient high-quality 3D motion data, current approaches still face challenges in generating fine-grained motions from unseen audio prompts. Therefore, we propose a novel framework to generate vivid and diverse gestures based on zero-shot human speech.

**Mixture-of-Experts.** Mixture-of-Experts (MoE) refers to combining the strengths of multiple expert models to improve model generalization performance (Fedus et al., 2022; Jacobs et al., 1991; Shazeer et al., 2017). Recently, MoE has been extensively applied to various research areas (Gale et al., 2023; Pini et al., 2023), demonstrating their versatility and effectiveness. In computer vision,

Table 1: Statistical comparison of our **GES-X** with existing ones. The dotted line separates whether the posture in the dataset is built based on the mesh. Among meshed whole body co-speech gesture datasets, the scale of our GES-X is 15× larger than the existing ones (*i.e.*BEAT2).

| Dataset | Duration (hours) | Attributes | | | | | | | Joint Annotation |
|---|---|---|---|---|---|---|---|---|---|
| | | Speakers | Facial | Mesh | Phoneme | Text | Body | Hand | |
| Trinity (Ferstl & McDonnell, 2018)$_{IVA}$ | 4 | 1 | ✗ | ✗ | ✗ | ✓ | 24 | 38 | mo-cap |
| TED (Yoon et al., 2020)$_{TOG}$ | 106.1 | 1,766 | ✗ | ✗ | ✗ | ✓ | 9 | ✗ | pseudo |
| SCG (Habibie et al., 2021)$_{CVPR}$ | 33 | 6 | ✗ | ✗ | ✗ | ✗ | 14 | 24 | pseudo |
| TED-Ex (Liu et al., 2022b)$_{CVPR}$ | 100.8 | 1,764 | ✗ | ✗ | ✗ | ✓ | 13 | 30 | pseudo |
| ZeroEGGS (Ghorbani et al., 2023)$_{CGF}$ | 2 | 1 | ✗ | ✗ | ✗ | ✓ | 27 | 48 | mo-cap |
| BEAT (Liu et al., 2022a)$_{ECCV}$ | 35 | 30 | ✓ | ✗ | ✓ | ✓ | 27 | 48 | mo-cap |
| TalkSHOW (Yi et al., 2023)$_{CVPR}$ | 26.9 | 4 | ✓ | ✓ | ✗ | ✗ | 24 | 30 | pseudo |
| BEAT2 (Liu et al., 2024a)$_{CVPR}$ | 27 | 25 | ✓ | ✓ | ✓ | ✓ | 24 | 30 | mo-cap |
| **GES-X (ours)** | **450** | **4,370** | ✓ | ✓ | ✓ | ✓ | **24** | **30** | pseudo |

researchers employ the MoE paradigm to facilitate the multi-modal alignment tasks (Feng et al., 2023; Wang et al., 2023). Concretely, Shen *et. al* (Shen et al., 2023b) specifically investigates the scalability of MoE in vision-language models and showcases its potential to outperform dense models with equivalent computational cost. Regarding the human motion task, Liang *et. al* (Liang et al., 2024) propose a mixture-of-controllers mechanism that adaptively recognizes various ranges of the sub-motions with the text-token-specific experts, resulting in significant improvement on the text2motion research. Moreover, we notice that Mixture-of-Modality-Experts achieve promising performance in long-sequence modeling tasks (Liu et al., 2023b; Puigcerver et al., 2023; Shen et al., 2023a; Zhang et al., 2018). Motivated by this, we introduce Mixture-of-Gesture-Experts in our framework to enhance long-sequence gesture generation upon human speech guidance.

## 3 PROPOSED METHOD

### 3.1 PROBLEM FORMULATION

With the specifically designed generation framework, our goal is to synthesize vivid and diverse 3D human gestures $X = \{x_1, ..., x_N\}$ of the upper body through the given unseen continuous human speech audio $A = \{a_1, ..., a_N\}$ as input. Here, $N$ denotes the number of the generated human postures coordinated with speech audio $A$. We leverage $J$ joints with 3D representation to indicate each pose $x_i$. Unlike the previous methods (Liu et al., 2024a; 2022a; Yi et al., 2023; Liu et al., 2024b) that either utilize the text transcripts or speaker ID embedding as auxiliary input, our CoCoGesture adopts only the human speech as model inputs. It should be noted this single modality input fashion significantly facilitates the unseen speech-conditioned co-speech gesture generation. Our overall workflow is displayed in Figure 3.

### 3.2 GESTURE DIFFUSION MODEL PRE-TRAINING

**Large-scale Co-speech Gesture Dataset.**  To ensure the generalization of our pre-trained transformer diffusion model, we newly collect a large-scale high-quality 3D meshed whole-body co-speech gesture dataset, dubbed GES-X. In particular, we first leverage the advanced 3D pose estimator Pymaf-X (Zhang et al., 2023a) to obtain the meshed whole-body parameters upon SMPL-X (Pavlakos et al., 2019). The original raw data is collected from about $4.3$K talk show videos including different stances (*i.e.*, standing or sitting). After data processing[1], our GES-X dataset contains more than $40$M gesture frames. To the best of our knowledge, this is the largest-scale whole-body meshed 3D co-speech gesture dataset, whose duration is 15x the current largest one, as reported in Table 1.

Specifically, the acquired human postures are represented as the unified standard SMPL (Loper et al., 2023) body model accompanied by the MANO (Boukhayma et al., 2019) hand model. The facial expression is presented in FLAME (Li et al., 2017) face model. Meanwhile, we leverage the powerful speech recognition model WhisperX (Bain et al., 2023) to gain accurate word-level text transcripts and linguistics phoneme (Studdert-Kennedy, 1987) aligned with the extracted motion dynamics. In this manner, our GES-X not only facilitates the research on co-speech gesture generation

---

[1]Please refer to supplementary material for more details.

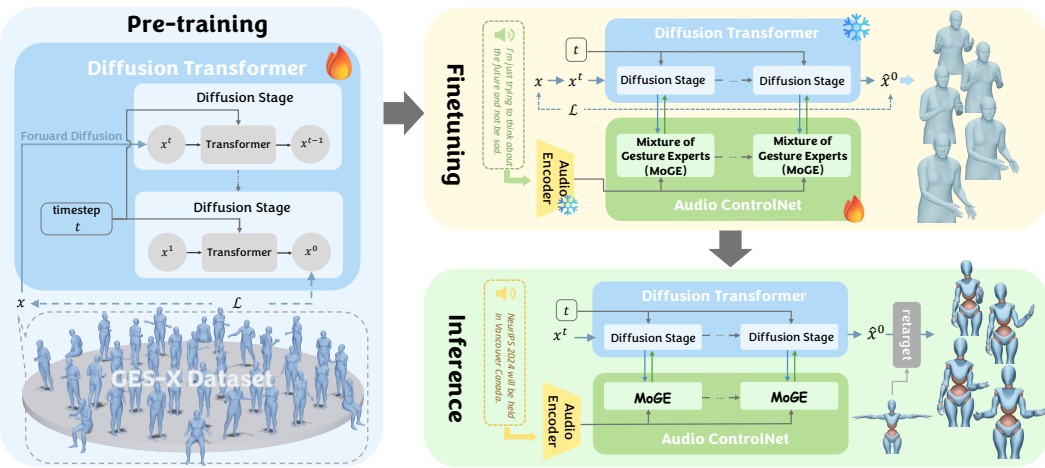

Figure 3: The overview of our CoCoGesture. In the **Pre-training**, we first pre-train a large uncondi-
tional diffusion model upon our large-scale GES-X dataset as the gesture expert. The **Finetuning**
stage incorporates audio signal as gesture generation guidance. In the **Inference** stage, our CoCoGes-
ture can generate vivid and diverse 3D co-speech gestures from unseen zero-shot human speeches.

but also supports various other human avatar creation tasks, *e.g.*, talking face (Tian et al., 2024),
human behavior analysis (Qi et al., 2023b). Along with this large-scale dataset, the pretraining of the
unconditional diffusion model is greatly enhanced with generalization and vividness.

**Model Scaling-up & Pre-training**   Inspired by (Guo et al., 2022; Liang et al., 2024), we formulate
the popular diffusion transformer (DiT (Peebles & Xie, 2023)) as our model backbone owing to the
scalability and excellent compatibility of large-scale training data. Here, similar to the foundation
model stable diffusion (Rombach et al., 2022), we scale up the original DiT from $120M$ to $1B$ with
different layers and latent dimensions, enabling learning massive gesture features so as to apply to
different downstream applications. During training, we enforce our denoiser to produce continuous
human motions given the diffusion time step $t$ and noised postures $x^t$. The denoising processing is
constrained by the simple objective:

$$\mathcal{L}_{simple} = \mathbb{E}_{x,t,\epsilon} \left[ \left\| x - \mathcal{D}_u(x^t, t) \right\|_2^2 \right], \tag{1}$$

where $\mathcal{D}_u$ is our unconditional denoiser, $\epsilon \sim \mathcal{N}(\mathbf{0}, \mathbf{I})$ is the added random Gaussian noise, $x^t = x + \sigma_t \epsilon$ is the gradually noise adding process at step $t$. $\sigma_t \in (0, 1)$ is the constant hper-parameter.
Moreover, we follow the setting of (Tevet et al., 2022b; Guo et al., 2022) to leverage the velocity loss
$\mathcal{L}_{vel}$ and foot contact loss $\mathcal{L}_{foot}$ for improving generated results more smoothness and physically
reasonable. To this end, the overall objective is

$$\mathcal{L}_{total} = \lambda_{simple}\mathcal{L}_{simple} + \mathcal{L}_{vel} + \mathcal{L}_{foot}, \tag{2}$$

where $\lambda_{simple}$ is trade-off weight coefficients.

### 3.3 AUDIO CONTROLNET FINETUNE

In the finetuning phase, we intend to incorporate the audio condition $A$ into the pre-trained gesture
model. Inspired by text2image ControlNet (Zhang et al., 2023b), we introduce an audio ControlNet
consisting of the trainable copy of the unconditional diffusion model and a novel proposed Mixture-
of-Gesture-Experts (**MoGE**) block, as shown in Figure 4. The frozen pre-trained model serves as
a strong gesture expert and the MoGE blocks follow a trainable copy to produce the temporally
coordinated joint embedding of the audio signal and gesture features. Then the joint embedding is
adaptively added to the denoised motion features of the next layer through a novel routing mechanism.

**Mixture-of-Gesture-Experts.**   Inspired by MoE (Zhu et al., 2024; Yu et al., 2024; Shazeer et al.,
2017), the key insight of the MoGE is adaptively fusing the information from the gesture expert (*i.e.*,
pre-trained model) and the speech audio expert (*i.e.*, audio encoder), thereby the generated gestures

preserving temporal consistent with speech rhythms. To enhance the sequence-aware correspondence of the fused features, we first leverage the audio embedding $\mathbf{f}^a$ as the query $Q$ to match the key feature $K$ and values features $V$ belonging motion embeddings $\mathrm{f}_l^{x''}$ via cross-attention mechanism:

$$Q_l = \mathbf{f}^a \mathbf{W}_l, K_l = \mathrm{f}_l^{x''} \mathbf{W}_l, V_l = \mathrm{f}_l^{x''} \mathbf{W}_l. \tag{3}$$

Here, $l$ represents the index of each attention layer, and $\mathbf{W}$ denotes the projection matrix. Once we obtain these fused trainable features $\mathbf{f}^{train}$, we adopt an adaptive instance normalization (Ada-IN) layer conditioned on audio features to further boost $\mathbf{f}^{train}$. Then, we utilize a learnable routing adaptor to combine the output of the gesture expert and trainable copy branch. To be specific, we leverage the output of the frozen original last layer as motion guidance representation to indicate the soft weight. By doing so, we derive the blending process as follows

$$\mathbf{f}_{l+1}^x = \mathbf{R}_l \odot \mathrm{f}_l^{x'} + (1 - \mathbf{R}_l) \odot \mathbf{f}_l^{train},$$
$$\mathbf{R}_l = Softmax(\mathbf{W}_{R,l} \otimes \mathrm{f}_l^x), \tag{4}$$

where $\mathbf{R}$ is the learnable router, $\mathbf{W}_{R,l}$ denotes the weight matrix, $\odot$ indicates the Hadamard product and $\otimes$ indicates matrix multiplication. Afterward, we exploit the zero-initialized convolution layers to ensure the audio condition in the trainable copy branch cannot be impacted by the harmful noise.

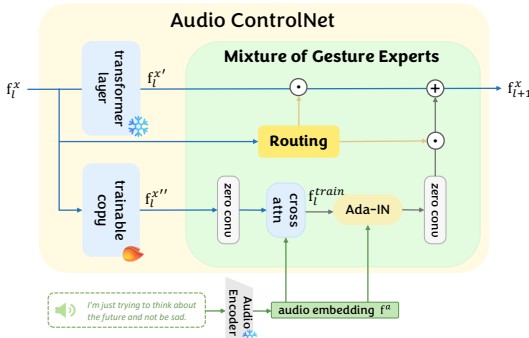

Figure 4: Details of our proposed Mixture-of-Gesture-Experts (MoGE) block. The pre-trained transformer layer is frozen and serves as the gesture expert, while the audio embedding is extracted from the audio expert.

**Training and Inference.** During the training, we leverage the same loss function in Eq. 2 to constrain the trainable conditional denoiser parameters. In the inference, we utilize the classifier-free guidance unconditional denoiser and audio-conditioned one $\mathcal{D}_a$:

$$\hat{\mathrm{x}}^{(0)} = s \cdot \mathcal{D}_a(\mathrm{x}^{(t)}, t, a) + (1 - s) \cdot \mathcal{D}_u(\mathrm{x}^{(t)}, t), \tag{5}$$

where $\hat{\mathrm{x}}^{(0)}$ denotes the denoised gesture motions, and $s$ is the set as $4.0$ in practice.

## 4 EXPERIMENTS

### 4.1 EXPERIMENTAL SETTING AND DATASETS

**Implementation Details.** In the pretraining phase, we set $\lambda_{simple} = 10$, empirically. The total diffusion time step is $1,000$ with the cosine noisy schedule (Nichol & Dhariwal, 2021). The initial learning rate is set as $1 \times 10^{-4}$ with AdamW optimizer. Our model is trained on 8 NVIDIA H800 GPUs with a batch size of 256. The total training process takes 100 epochs, accounting for one week of the largest model version within 1B parameters. We provide three-version models with different architectures and parameters to explore the dependence of performance on model size.

During the finetuning stage, the audio signal is processed to mel-spectrograms with FFT window size $1,024$, and hop length $512$. Similar to (Liu et al., 2022b; Qi et al., 2023a; 2024), we take an advanced speech recognizer (Chung et al., 2020) as the audio encoder. We train the audio ControlNet with a batch size of 128 for 100 epochs. The initial learning rate is set as $1 \times 10^{-5}$. We take the DDIM (Song et al., 2020) sampling strategy within 25 denoising timesteps during inference. Temporally, our CoCoGesture synthesizes the 10-second gesture motions including 43 upper joints (*i.e.*13 body joints + 30 hand joints) in practice. Each joint is converted to the 6D rotation representation (Zhou et al.) for better modeling in the experiments.

**GES-X Dataset.** We newly propose a large-scale co-speech gesture dataset, dubbed GES-X, to train our unconditional diffusion model. Firstly, we leverage 16 NVIDIA RTX 4090 GPUs to extract the 3D human poses from downloaded in-the-wild $4,370$ talk show videos. This process takes ***more than***

Table 2: Comparison with the state-of-the-art counterparts on BEAT2 and TalkSHOW datasets. ↑ means the higher the better, and ↓ indicates the lower the better. "-" denotes that the method cannot be applied to the TalkSHOW dataset due to the lack of text transcripts. The term "zero-shot" implies that the dataset contains unseen human voices.

| Methods | BEAT2 (Liu et al., 2024a) | | | TalkSHOW (Yi et al., 2023) (zero-shot) | | |
|---|---|---|---|---|---|---|
| | FGD ↓ | Diversity ↑ | BA ↑ | FGD ↓ | Diversity ↑ | BA ↑ |
| Trimodal (Yoon et al., 2020)$_{TOG}$ | 13.05 | 33.54 | 0.75 | - | - | - |
| HA2G (Liu et al., 2022b)$_{CVPR}$ | 9.37 | 45.81 | 0.76 | 15.25 | 58.41 | 0.65 |
| CAMN (Liu et al., 2022a)$_{ECCV}$ | 7.12 | 44.02 | 0.82 | - | - | - |
| TalkSHOW (Yi et al., 2023)$_{CVPR}$ | 10.59 | 45.23 | 0.79 | 16.41 | 57.30 | 0.64 |
| DiffuGesture (Zhu et al., 2023)$_{CVPR}$ | 11.82 | 48.53 | 0.81 | 17.03 | 50.52 | 0.72 |
| ProbTalk (Liu et al., 2024b)$_{CVPR}$ | 6.06 | 66.03 | 0.82 | 11.18 | 65.95 | 0.78 |
| EMAGE (Liu et al., 2024a)$_{CVPR}$ | 4.09 | 69.70 | 0.85 | - | - | - |
| **CoCoGesture (ours)** | **3.92** | **70.47** | **0.87** | **9.62** | **69.10** | **0.83** |

*one month*, acquiring more than 88 million raw frames. After filtering the unreasonable gestures, we obtain 40 million high-quality postures. Then, we resample the FPS as 15, thereby the total generated gesture frames are 150 in a sequence. Finally, we obtain the $100, 162$ motion clips with corresponding audio/text transcripts/phonemes.

**BEAT2 and TalkSHOW Datasets.** To fully verify the generalization and effectiveness of our pr-trained model, we adopt two meshed datasets BEAT2 (Liu et al., 2024a) and TalkSHOW (Yi et al., 2023) in the evaluation phases. BEAT2 contains 3D meshed whole-body postures with multi-modality information such as speaker ID and text transcripts. The content of the speech is based on 25 speakers' answers to predefined questions. All the instances in BEAT2 are standing postures collected by the motion-capture system. In the TalkSHOW dataset, only sitting postures with 4 speakers are collected by 3D pose estimator from in-the-wild talk show videos. It is noted that the TalkSHOW dataset does not provide text transcript annotation.

**Evaluation Metrics.** To fully evaluate the realism and diversity of the generated co-speech gestures, we introduce various metrics:

- **FGD**: Fréchet Gesture Distance (FGD) (Yoon et al., 2020) is leveraged to measure the distribution distance between the motions of real ones and generated ones.

- **BA**: Beat Alignment Score (BA) (Liu et al., 2022a;b) measures whether the generated human motions are rhythmically aligned with the speech beat.

- **Diversity**: Similar to (Liu et al., 2022b; Zhu et al., 2023; Qi et al., 2024), the same feature extractor is exploited to acquire feature embeddings of the synthesized gestures. We leverage the average distance between 500 randomly assembled pairs to indicate the diversity score.

## 4.2 QUANTITATIVE RESULTS

**Comparisons with the State-of-the-art.** To fully verify the effectiveness of our method, we compare our CoCoGesture framework with various state-of-the-art counterparts: Trimodal (Yoon et al., 2020), HA2G (Liu et al., 2022b), CAMN (Liu et al., 2022a), TalkSHOW (Yi et al., 2023), DiffuGesture (Zhu et al., 2023), ProbTalk (Liu et al., 2024b) and EMAGE (Liu et al., 2024a). For a fair comparison, all the models are implemented by the source code released by the authors. We adopt GES-X in the finetuning stage to train our audio ControlNet. Then, we exploit both BEAT2 and TalkSHOW as testing sets. As for all the other counterparts, we adopt only the BEAT2 as the training set. The TalkSHOW serves as the out-of-domain testing dataset, measuring the comparison of the zero-shot ability. Since the TalkSHOW dataset does not provide the text transcript, it cannot be used by some competitors (Yoon et al., 2020; Liu et al., 2022a; 2024a) that rely on text.

As reported in Table 2, our framework achieves the best results on both datasets. We observe that both EMAGE and ours generate high-quality results in the FGD metric on the BEAT2 dataset. However, different from EMAGE trained on BEAT2, our CoCoGesture is directly tested on this dataset. Meanwhile, since our method only depends on the audio signal input, we can easily apply it to another dataset. In terms of diversity score, our classifier-free inference strategy enables diverse

Table 3: Ablation study on model scale and pre-training setting. ‡ denotes without pre-training stage.

| Model | $n_{layers}$ | $d_{model}$ | $n_{heads}$ | $d_{heads}$ | Parms | BEAT2 (Liu et al., 2024a) | | |
|---|---|---|---|---|---|---|---|---|
| | | | | | | FGD ↓ | Diversity ↑ | BA ↑ |
| CoCoGesture-Base | 25 | 512 | 8 | 128 | 120M | 6.00 | 52.73 | 0.81 |
| CoCoGesture-Medium | 25 | 1024 | 16 | 128 | 480M | 4.96 | 57.75 | 0.83 |
| CoCoGesture-Large ‡ | 50 | 1024 | 16 | 128 | 1B | 4.30 | 68.33 | 0.85 |
| **CoCoGesture-Large** | 50 | 1024 | 16 | 128 | 1B | **3.92** | **70.47** | **0.87** |

gestures while preserving the authority and vividness of the results. Considering the zero-shot inference, our approach outperforms all the counterparts by a large margin. Remarkably, on the TalkSHOW dataset, our CoCoGesture reduces FGD by a significant amount of $16.22\%$ over the sub-optimal counterparts. The better performance demonstrates our model's superior generalization ability, verifying our insight on pre-training and finetune strategy.

**Ablation Study.** To further evaluate the effectiveness of our proposed framework, we conduct a series of ablation studies of different components and training strategies as variations.

**Effects on Model scale & Pre-training:** To investigate the impact of the model scale and pre-training stage, we conduct the ablation study on the BEAT2 dataset, as reported in Table 3. We design three model variants with different architectures. Here, $n_{layers}$ is the total transformer layers, $d_{model}$ denotes dimension of latent vectors, $n_{heads}$ means number of attention heads, $d_{heads}$ indicates the dimension of each attention head. It is observed that our model performance is gradually improved with model scaling up. This aligns our insight on larger models to learn massive gesture manifold. It is noticed that without pre-training, the model achieves lower performance. This suggests that pre-training on our GES-X dataset is effective in improving model generalization ability.

**Effects of the MoGE Block:** To fully analyze the effectiveness of our proposed Mixture-of-Gesture-Experts (MoGE) blocks, we conduct the ablation study through detailed components. As reported in Table 4, we demonstrate the exclusion of cross-attention and routing mechanisms respectively from our full large model version leads to performance degradation. To be specific, the cross-attention module effectively models the dependency of audio signals with generated results, thus implementation without it leads to worse performance in all the met-

Table 4: Ablation study of MoGE block on BEAT2 dataset.

| Methods | BEAT2 (Liu et al., 2024a) | | |
|---|---|---|---|
| | FGD ↓ | Diversity ↑ | BA ↑ |
| w/o Cross-attn | 4.79 | 62.48 | 0.86 |
| w/o Routing | 4.28 | 67.14 | 0.79 |
| **CoCoGesture (full)** | **3.92** | **70.47** | **0.87** |

rics. Meanwhile, the exclusion of the routing mechanisms results in an obvious decrease in the BA score. This demonstrates that our routing mechanism significantly enhances the temporal coherency between the audio embeddings *w.r.t.* gesture features, thus producing vivid and coherency gestures.

## 4.3 QUALITATIVE EVALUATION

**Visualization:** To fully demonstrate the superior performance of our CoCoGesture framework, we show the visualized key frames synthesized by ours compared with various counterparts on BEAT2 and TalkSHOW datasets, respectively. As shown in Figure 5, our method displays vivid and diverse gestures against others. In particular, we observe the Trimodal tends to synthesize unreasonable and stiff results (*e.g.*, the red rectangle in the BEAT2 dataset). Although the HA2G and EMAGE can generate the natural upper body postures, we find that their body movements are of limited dynamics (*e.g.*, the blue rectangle in the BEAT2 dataset). In terms of the zero-shot inference in the TalkSHOW dataset, both DiffuGesture and our method produce reasonable gestures. However, the results generated by DiffuGesture are misaligned with the input audio. This may be caused by the limited word corpus of the BEAT2 dataset restricting the generalization of the model. In contrast, our method can synthesize the vivid and synchronous co-speech gestures (*e.g.*, the arms become lifting while the hands stretch out). This highly aligns with our motivation about the model generalization improved by pre-training on our large-scale dataset GES-X. For more demo results please refer to our anonymous website: *https://anonymous.4open.science/w/GES-X/*.

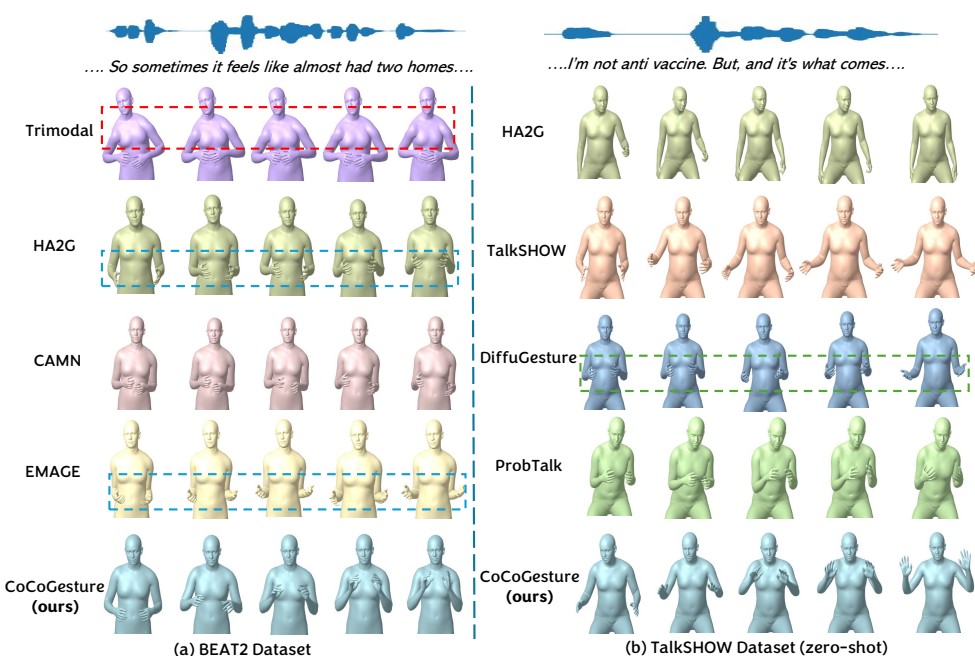

Figure 5: Visualization of our generated 3D co-speech gestures against various state-of-the-art methods. The samples on the left are from BEAT2, and the samples on the right are from TalkSHOW.

**User Study:** To further analyze the quality of results synthesized by various counterparts and ours, we conduct a user study by inviting 15 volunteers. The statistical mean results are reported in Figure 6. All the volunteers are recruited anonymously from schools with different majors. Each participant is required to rate the randomly selected visualization videos from 0 (worst) to 5 (best) in terms of naturalness, smoothness, and speech-gesture coherency. Our CoCoGesture framework demonstrates the best performance among all the competitors. Especially, in terms of smoothness and speech-gesture coherency, our method outperforms others with noticeable improvements, verifying the effectiveness of our Mixture-of-Gesture-Expert.

## 5    CONCULSION

In this paper, we propose **CoCoGesture** to generate vivid and diverse co-speech 3D gestures from in-the-wild zero-shot human speech. To fulfill this goal, we first newly collect a large-scale dataset that contains more than 40M high-quality 3D meshed postures across 4.3K speakers from in-the-wild talk show videos. Along with this dataset, we pre-train a large generalizable diffusion model to be our gesture expert in

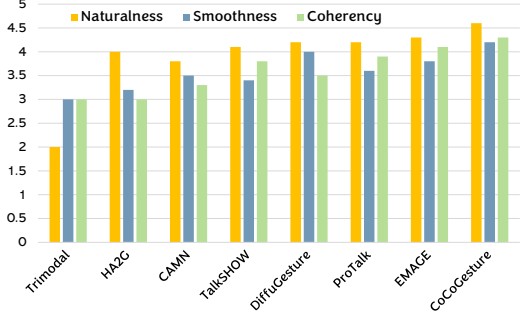

Figure 6: User study on gesture naturalness, motion smoothness, and speech-gesture coherency.

the first stage. To incorporate human speech as guidance, we further propose a novel audio ControlNet that adaptively fuses the audio embeddings and the motion clues from the pre-trained gesture expert. Extensive experiments conducted on two out-of-domain datasets show the superiority of our model.

**Limitation:** Our framework only takes the audio signal as model input to generate gestures. It might be possible that our model produces emotionally insensitive cases (*e.g.*, moving faster or more intensely when angry or happy). Meanwhile, the automated pose extraction and speech techniques may have an impact on the datasets we newly collect, despite the huge effort we put into data clean filtering and processing. In future works, we will incorporate our model with emotional conditions and investigate more stable data processing techniques to improve the quality of generated gestures.

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

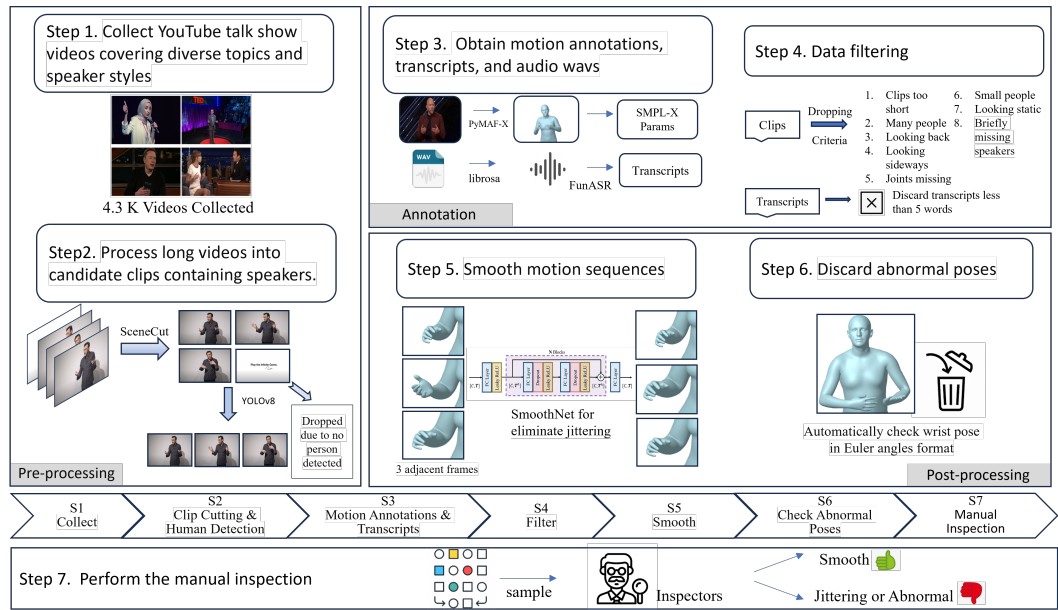

Figure 7: The overall workflow of our dataset construction. The talk show videos are processed to obtain high-quality postures through advanced automatic technologies and expert proofreading.

# A    SUPPLEMENTARY MATERIAL

To demonstrate the effectiveness of our data construction techniques and the proposed method of coherent co-speech gesture generation, we further elaborate on the detailed data synthesis and vision perception in the supplementary material.

## A.1    DATASET

### A.1.1    CONSTRUCTION OF OUR GES-X

In this section, we detail the overall pipeline for creating GES-X, a large-scale dataset that contains over 40M co-speech gesture frames. The whole procedure consists of four folds: internet video collection, motion annotation, post-processing, and manual inspection, as summarised in Figure 7.

**Internet Videos Collection (Step 1&2):** Acquiring the paired speech-gesture 3D data via motion capture system is expensive and labor-consuming. Consequently, some previous works (Liu et al., 2022b; Yoon et al., 2020; 2019; Qi et al., 2023a; 2024; Yi et al., 2023) leverage in-the-wild talk show videos as the source to extract 3D postures via advanced pose estimator. Following this fashion, we intend to obtain large-scale co-speech 3D gestures from YouTube talk show videos covering diverse topics and speaker styles. We obtain 4,370 videos and their corresponding text transcripts. Given the substantial volume of our video data, we employ PySceneDetect to segment lengthy videos into clips. YOLOv8 is also used for human detection, discarding clips that do not show a person within the first 30 frames. These processes allow us to obtain potential clips containing speakers, with an average duration of 9.85 seconds of each.

**Motion Annotation (Step 3&4):** Here, we employ SMPL-X (Pavlakos et al., 2019) to represent whole-body poses, a widely 3D human representation standard adopted in various downstream tasks. Then, we exploit the advanced pose estimator PyMAF-X (Zhang et al., 2023a) to extract high-quality 3D postures including body poses, subtle fingers, shapes, and expressions of the speakers. For audio processing, we use FunASR (Gao et al., 2023) with the Whisper-large-v3 model to generate transcripts. We then apply eight criteria to filter the clips and motion annotations: *clips that are too short, contain multiple people, involve looking back or sideways, have missing joints, show small or static individuals, or briefly miss the speakers*. Additionally, transcripts with fewer than five words are discarded, though the corresponding video clips are retained to increase the data scale for certain audio-to-gesture tasks.

**Post-Processing (Step 5&6):** Once we obtain a large amount of raw pose sequences, we conduct the post-processing to boost the quality of our data. Specifically, we visualize the motion sequences with render mesh vertices and observe there are some temporal jittering issues. These jitters usually result from heavy occlusion, truncation, and motion blur caused by changes in camera angles and large-scale human movements of speakers. To address this, similar to CLIFF (Li et al., 2022), we utilize SmoothNet (Zeng et al., 2022) for temporal smoothing and jitter motion refinement. In practice, through manual review, we notice that SmoothNet effectively produces cleaner and more reliable motion sequences without sacrificing the diversity of postures. Despite that, given the frequent extreme variations in camera angles, speaker poses, and lighting in talk show videos, some inaccurate pose estimations from PyMAF-X are inevitable. Therefore, we leverage an automatic abnormal pose detection method to further improve the pose quality. By representing the arm poses as Euler angles using the $x$, $y$, and $z$ convention, based on findings from (Pavlakos et al., 2019), we focus particularly on the poses of the wrists. Once the wrist poses exceed 150 degrees on any axis or if the pose changes by more than 25 degrees between adjacent frames (at 15 fps), we discard these abnormal postures surrounding 150 frames.

**Manual Inspection (Step 7):** Finally, we perform the manual review for the processed clips with a uniform ratio of 10:1. In particular, we follow the order of scenecut and sample one clip from every ten groups of clips. Since these 10 clips typically originate from the same video, making this assumption reasonably valid. For all clips, we divide them into ten groups for ten inspectors to manually review. These inspectors evaluate the visualizations based on obtained SMPL-X parameters to determine whether they are smooth, jittering, or abnormal. If the motion sequences appear jittering or abnormal, the entire group of ten clips from which the sample originated is discarded. Through meticulous evaluation and significant effort, the quality of our GES-X is greatly ensured.

**Text Transcript and Phonme Alignment:** To acquire accurate semantic annotations from speech, we transcribe audio files to extract text, phonemes, and their corresponding timestamps. Specifically, we utilize WhisperX (Bain et al., 2023) as our transcription tool, which employs pyannote (Bredin, 2023) for speaker diarization and the Whisper (Radford et al., 2023) model for automatic speech recognition (ASR). This tool incorporates a VAD Cut & Merge strategy to address the issue of inaccurate timestamp predictions in long audio. We configure the system to recognize only one speaker and utilize the Whisper Large V3 model for ASR. This approach splits long audio into segments, each with its corresponding text. Subsequently, all data and labels are manually reviewed by skilled human annotators. Finally, we apply the verified transcriptions and segment results to perform Forced Phoneme Alignment using the Montreal Forced Aligner (McAuliffe et al., 2017) to accurately label all phonemes and their respective timestamps.

### A.1.2   BEAT2 & TALKSHOW DATASETS

Similar to our GES-X, we first resample the BEAT2 and TalkSHOW datasets with the FPS 15. Then, we divide datasets into 10s clips. Finally, we obtain $35, 758$ clips in BEAT2 and $9, 629$ in TalkSHOW. We follow the convention of (Liu et al., 2024a) to split the train/validation/test with the proportion of $85\%$, $7.5\%$, and $7.5\%$ of both datasets.

### A.2   ADDITIONAL EXPERIMENTS

### A.2.1   METRIC CALCULATION DETAILS

Inspired by (Yoon et al., 2020; Liu et al., 2022b), we leverage the FGD to evaluate whether the generated gestures preserve realism with the ground truth in the perceptive of distribution. We first pre-train an auto-encoder as the feature extractor. Then the FGD is calculated among the latent vectors belonging to sequential prediction and ground truth, respectively. The dimension of the latent vector is 128, similar to (Yoon et al., 2020; Liu et al., 2022b).

### A.2.2   DISCUSSION OF EXPERIMENTAL SETTING

In our experiments, we only take human audio as a condition to guide the gesture generation. Although current speech-to-text methods can provide high-quality results, it requires an additional module to obtain word-level transcripts with accurate timestamps before modeling gestures from human speech. Meanwhile, during our pretraining phases, there are more than 4.3k speaker identities. In this fashion,

it is difficult to model the speaker's characteristics. In contrast, our method directly generates the gestures from speech signals. In this universal manner, our model is more practical in real sence applications (e.g., outdoor background noise may have a serious impact on speech-to-text). Therefore, similar to (Yi et al., 2023; Liu et al., 2024b; Zhu et al., 2023), our setting of directly generating gestures from speech audios without textual information is one of the common methodology streams in the community.

### A.2.3 Additional Ablation Results

We further conduct experiments to train our CoCoGesture on the BEAT2 dataset (denoted as CoCoGesture*). Our method attains the best performance against all the counterparts, which highly demonstrates the effectiveness of our proposed CoCoGesture framework. Although the FGD of our framework pre-trained on the GES-X dataset (denoted by †) is slightly worse than the one trained on BEAT2 due to cross-dataset evaluation, it still achieves better results than other competitors.

Table 5: Ablation study of pre-training on BEAT2 dataset.

| Methods | BEAT2 (Liu et al., 2024a) | | |
| --- | --- | --- | --- |
| | FGD ↓ | Diversity ↑ | BA ↑ |
| **CoCoGesture*** | **3.66** | **71.08** | **0.87** |
| CoCoGesture† | 3.92 | 70.47 | 0.87 |

### A.2.4 User Study Details

During the user study, we utilize eight models to randomly generate demo videos in each of the BEAT2 and TalkSHOW datasets. For each method, we randomly generate two demo videos from two datasets. For those that can be performed on the Talkshow dataset, the generated results are guaranteed to come from both datasets. Therefore, each participant needs to respond to 16 samples from eight methods. Then, all the volunteer students are requested to rate all videos without any hint about which model produces this video. The higher score means the better results. 5 points means that the video meets the audience's requirements perfectly. 0 points indicates that the video is totally unacceptable. To ensure fairness, each demo video is played on a PPT slide with a blank background. When all students have completed the grading, their results will be collected anonymously and the average score will be calculated and announced. For each sample, the participants are allowed to rate only after watching the entire video. To ensure that participants will not have biased results due to recency bias, we invite participants to take the test at different periods and not strictly limit the test duration. Participants can watch each video repeatedly. We double-check the rating results by randomly selecting 60% of participants to redo the same test one week later, and there are no significant changes to the final results.

### A.2.5 Additional Visualization Results

Here, we provide more visualized results of our CoCoGesture framework and other counterparts in the anonymous website: *https://anonymous.4open.science/w/GES-X/*. Moreover, to fully demonstrate the effectiveness of our proposed components and different model scales, we visualize the key frames of the generated results in Figure 8 and Figure 9.

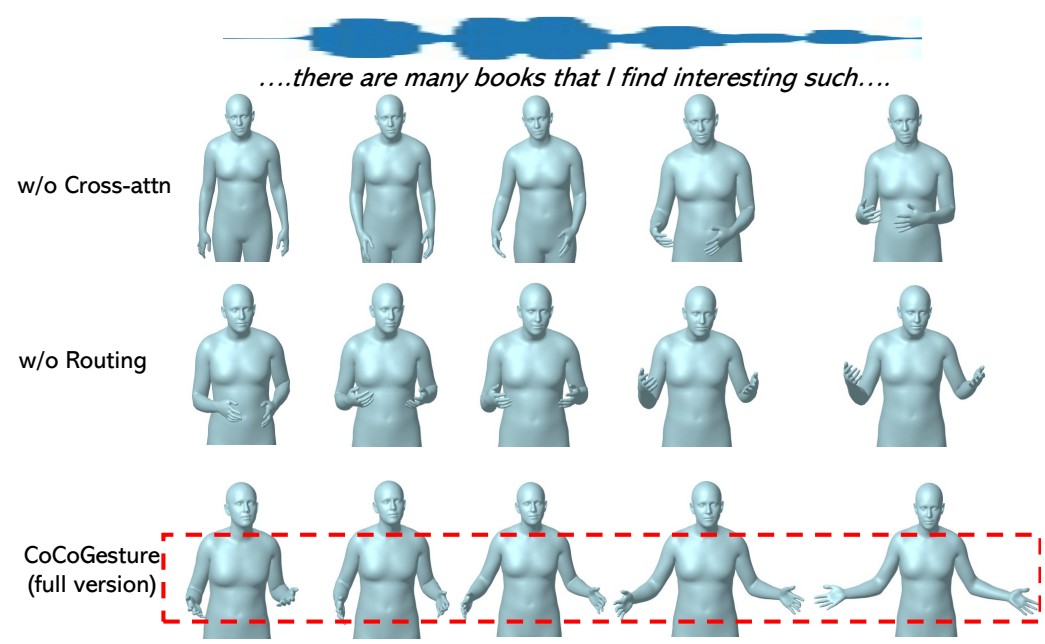

Figure 8: Visual comparisons of ablation study on BEAT2. We show the key frames of the generated motions given the human speech. Best viewed on screen.

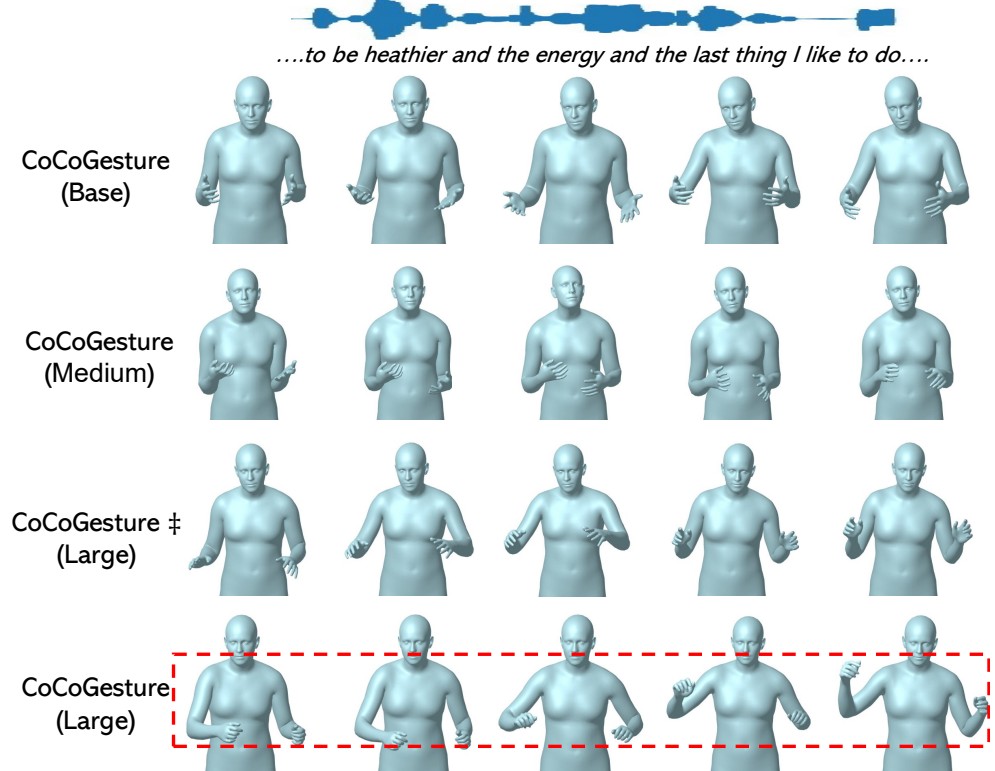

Figure 9: Visual comparisons of ablation study on BEAT2. We show the key frames of the generated motions given the human speech. Best viewed on screen.

