# OpenReview forum: "$\textbf{CoCoGesture}$: Towards Coherent Co-speech 3D Gesture Generation in the Wild"
_ICLR.cc/2025/Conference — ICLR 2025 Conference Withdrawn Submission_

### Official Review · Reviewer_Jz5Q · 2024-10-20

**Soundness:** 3
**Presentation:** 3
**Contribution:** 4
**Rating:** 6
**Confidence:** 5

**Summary:**

This paper focuses on **scaling up training** in the context of co-speech 3D upper-body gesture generation. Before this work, previous works trained models on medium-scale datasets of 26 to 100 hours. To achieve large-scale training, the authors contribute:

1. **Data Scaling**:
   The authors construct a large-scale pseudo-label dataset, GES-X, consisting of 450 hours of data. The dataset is processed using a seven-step data filtering procedure to ensure quality.

2. **Model Scaling**:
   They propose a framework that combines a Diffusion Transformer (DiT), a pretraining and fine-tuning schedule, and a Mixture-of-Gesture-Experts (MoGE) module. This method enables training models ranging in size from 120M to 1B parameters.

3. **Evaluation**:
   The pretrained models are evaluated by fine-tuning on the BEAT2 dataset and performing zero-shot inference on the TALKSHOW dataset. The results show that the proposed model, CoCoGesture, raise new SOTA results.

**Strengths:**

Overall, this is a solid work. The community lacks exploration of how to scale up the co-speech gesture generation task, and this work is, to the best of my knowledge, the first attempt to address this. It brings a series of insights and experimental results. Details are listed below:

1. **Dataset Contribution**:
   The proposed dataset and, more importantly, the detailed process for obtaining a scaled dataset will benefit the community. In addition to Talkshow [Yi et al., 2023], this paper discusses a seven-step data filtering procedure, which is critical for model performance. Typically, raw outputs from 3D SMPL-X estimation methods have jitter issues. This paper introduces SmoothNet [Zeng et al., 2022] to address this.

2. **Model Insights**:
   The proposed model adapts insights from general computer vision tasks, such as text-to-image and pose-to-image, to the motion generation domain, specifically for audio-to-gestures. The choice of Diffusion Transformer (DiT) is reasonable for training on large-scale data, and the ControlNet-like architecture and zero-convolution modules is appropriately applied.

3. **Mixture-of-Gesture-Experts (MoGE)**:
   The Mixture-of-Gesture-Experts (MoGE) plays a key role in scaling up this task. Since inference speed is crucial for real-world co-speech gesture generation applications, especially with large models (~1B parameters), training with full parameters and inferring with partial, selective parameters is a clever approach.

4. **Experiments**:
   The experiments are comprehensive. BEAT2 [Liu et al., 2024] and TALKSHOW [Yi et al., 2023] are commonly used 3D mocap and pseudo datasets for this task. On the TALKSHOW dataset, the zero-shot performance exceeds the models trained on the dataset, demonstrating the potential for real-world applications.

**Weaknesses:**

For the weaknesses, I have one suggestion about the lip region:

1. **Visualization of Lip Region**:
   As I understand, the main focus of this paper is upper body gestures, but the visualizations include lip motion. I suggest the authors add a note in the demo videos indicating that "our model focuses on body gestures" to avoid confusion regarding the scope of the work.

**Questions:**

One question as same as weakness will influence my opinions:

1. **Visualization of Lip Region**:
   As I understand, the main focus of this paper is upper body gestures, but the visualizations include lip motion. I suggest the authors add a note in the demo videos indicating that "our model focuses on body gestures" to avoid confusion regarding the scope of the work.

Two questions are additional suggestions, and unclear points, will not influence my score.

1. **Release of Raw and Filtered Dataset Versions**:
   Releasing both the raw and filtered versions of the dataset could have a broader impact, allowing users to explore additional tasks, such as training with noisy data. This would provide flexibility for various use cases in the community.

2. **Clarification on Sequence Length for Transformer Training**:
   Could you clarify the sequence length used during transformer training? For instance, were shorter clips like 64 frames used, or were longer sequences such as 512 frames utilized? If shorter sequences were used, how was longer motion generated? Was a method like split inference for each part with blending applied?

---

### Official Review · Reviewer_ZX9N · 2024-10-30

**Soundness:** 2
**Presentation:** 3
**Contribution:** 1
**Rating:** 3
**Confidence:** 4

**Summary:**

This paper introduces an audio2gesture model for co-speech upper-body human gesture synthesis. A new large-scale audio-gesture dataset is obtained by pseudo annotation of videos. A two-stage strategy is proposed to train the 1B large model for co-speech synthesis.

**Strengths:**

1. A large-scale audio-gesture dataset with an abundant number of speakers is collected.
2. Ablations are done for the proposed design choices.

**Weaknesses:**

1. The generated motion lacks movements, with little correspondence to the speech. The results look over-smoothed. I’m wondering if randomly switching to unmatched speech can make any noticeable difference since the speech and the generated motion have little correlation.
2. The comparison of model size is only done via metrics. How is the visual quality affected by the model size? Given that the best model still leads to over-smoothed gestures and fails to capture the speech characteristic, I strongly argue to provide visual quality comparison about the smaller models, as well as the other baselines with similar model size.

**Questions:**

1. Why divide the training into two stages? Since the GES-X dataset contains both audio and gestures, why not train the model end-to-end similar to previous methods? ControlNet can help achieve different tasks simultaneously by freezing the backbone models, however here the task is solely audio-to-gesture, why using this ControlNet approach instead of end-to-end training?
2. What’s the criteria of filtering out bad gestures in the data processing?
3. Why the other baselines are trained on BEAT2 but the proposed method is trained on GES-X? Why not train all methods on the same GES-X dataset?

---

### Official Review · Reviewer_3F3C · 2024-11-03

**Soundness:** 2
**Presentation:** 2
**Contribution:** 2
**Rating:** 3
**Confidence:** 4

**Summary:**

The paper propsoed a co-speech gesture generation from in-the-wild human speech incorporating the large 3D meshed whole-body human posture dataset. This is so far the largest dataset of its kind.  It contains more than 40M meshed posture instances across 4.3K speakers.
Besides this GES-X dataset, the paper also proposes a novel framework named CoCoGesture that leverages the Mixture-of-Gesture-Experts (MoGE) blocks to adapt various unseen audio signals.

The paper also borrowed a controlnet finetuning idea from the text2image. This includes an audio ControlNet consisting of the trainable copy of the unconditional diffusion model and a novel proposed Mixture- of-Gesture-Experts (MoGE) block to adapt various unseen audio signals. Hence the propose method is more robust towrads the zero-shot unseen audio senario.

**Strengths:**

The paper proposed the co-speech gesture generation dataset which is the largest of the kind, which defintiely benefits the community.
This is the largest-scale whole-body meshed 3D co-speech gesture dataset, whose duration is 15x the current largest one. The approach uses a series of state-of-the art moduel to collect data,  such as the detection with the  YOLOv8,  3D pose mesh estimation using the SMPL-X.

**Weaknesses:**

1. Besides the proposed dataset which is the strength, the weakness is it's methodology . The porposed method is limited in its novelty. Both controlnet and Mixture-of-Gesture-Experts are of-the-shelf and hence difficult to justify the paper's originaltiy and novlety merits.
Current form of the paper presentation deminishes the algorith novelty, while emphasizing the dataset.  This novelty is difficult to be accepted by the ICLR level of conference.

A few actionable items are suggested below :
(1) the authors are suggested to provide more technical details on the MoGE part of the section 3.3, especially the training and the inference.  (2) the authors are sugested to rearrange the paper structure, by highlighting the algortihm of the CocoGesture, with the pre-training as part of this framework.  (3) the authors are suggested to introduce the dataset as an additional asset to benefit the research community, not as the primary contribution (currently contribution item #1) for this techncial paper.

**Questions:**

1. The dataset adopts the gestures from the SMPL-X model for the 3D mesh. However SMPL-X itself also has bias on the 3D pose estimation, which the bias is from its samples of the training data. The paper omits such potential misalignment.
This is suggested to discuss how the discrepency might be happening due to the SMPL-X's own limitation.

2. It is unclear on how the audio ControlNet that incorporates the human voice as condition prompts to guide the gesture generation.
For instance the controlnet in the text2image modality uses additional image prompt such as the pose to guide the image generation, in which pose will be guiding the similar pose in the generated image. Here, the concept of audio controlnet is not clear and perhasp an analogy with the text2image controlnet should be drawn to demonstrate the originality of the audio Controlnet.
Some acitonable items are below:
(1) Since the ControlNet of the image generation focus on controlling signal to modify the U-net structure in its encoder blocks, the U-net structure is the image generation pipeline，shown as Figure 3. Reference [1].  If using contronet as a reference for the proposed "audio contronet" paradigm, it is suggested to resketch the Figure 4 following a suitable "U-net" scenario for the co-speech scenario.
This should include more details of how the trainable copy of the "audio controlnet" is actually control the freezing branch. (2) Figure 4 description in section 3.3 first paragaph is too simple. As a result, it is diffcult to follow each blocks' usage and significance of the "audio controlnet".  A more detailed block-by-block explanation is suggested.

Reference:
[1] Adding Conditional Control to Text-to-Image Diffusion Models  https://arxiv.org/pdf/2302.05543

---

### Note · Authors · 2024-11-13

I have read and agree with the venue's withdrawal policy on behalf of myself and my co-authors.